# Modeling Dynamic Missingness of Implicit Feedback for Recommendation

**Menghan Wang**
College of Computer Science,
Zhejiang University
wangmengh@zju.edu.cn

**Mingming Gong**
Department of Biomedical Informatics,
University of Pittsburgh
mig73@pitt.edu

**Xiaolin Zheng**[*]
College of Computer Science,
Zhejiang University
xlzheng@zju.edu.cn

**Kun Zhang**
Department of Philosophy,
Carnegie Mellon University
kunz1@cmu.edu

## Abstract

Implicit feedback is widely used in collaborative filtering methods for recommendation. It is well known that implicit feedback contains a large number of values that are *missing not at random* (MNAR); and the missing data is a mixture of negative and unknown feedback, making it difficult to learn users' negative preferences. Recent studies modeled *exposure*, a latent missingness variable which indicates whether an item is exposed to a user, to give each missing entry a confidence of being negative feedback. However, these studies use static models and ignore the information in temporal dependencies among items, which seems to be an essential underlying factor to subsequent missingness. To model and exploit the dynamics of missingness, we propose a latent variable named "*user intent*" to govern the temporal changes of item missingness, and a hidden Markov model to represent such a process. The resulting framework captures the dynamic item missingness and incorporate it into matrix factorization (MF) for recommendation. We also explore two types of constraints to achieve a more compact and interpretable representation of *user intents*. Experiments on real-world datasets demonstrate the superiority of our method against state-of-the-art recommender systems.

## 1   Introduction

Collaborative filtering methods based on implicit feedback (e.g., purchase records and browsing history) are widely used in recommender systems. Compared to explicit feedback (e.g., 1-5 star ratings), implicit feedback is more abundant and accessible in real-world applications. However, the missing data of implicit feedback also brings two challenges. First, the data is *missing not at random* (MNAR). Only positive feedback is collected in implicit feedback and all negative feedback is missing, leading to a severely biased dataset. Second, the missing data is a mixture of negative and unknown feedback; a missing entry may indicate the user either dislikes or does not know the item, which makes it hard to learn user's negative preferences. Several previous works [Hu et al., 2008, Marlin and Zemel, 2009] provided evidence that both ignoring missing data and treating all missing data as negative feedback will lead to biased recommendations.

A possible solution is to model the MNAR mechanism and treat the missing data properly. Several researchers have proposed various methods to address this issue. Popular methods [Hu et al., 2008,

---

[*]Corresponding author

Pan et al., 2008] are based on the uniformity assumption that assigns a uniform weight to degrade the importance of the missing data, assuming that each missing entry is equally likely to be negative feedback. This is a strong assumption and limits models' flexibility for real applications. Recently, researchers [Liang et al., 2016, Wang et al., 2018a] treated missing entries differently with the so-called *"exposure"* variables and achieved improved results. An *exposure* variable indicates whether or not an item is missing to a user. They make predictions in two steps: They first model *exposure* variables for each user to get the candidate items that are not missing and then recommend top-ranked items in the set of candidate items based on user preferences.

However, these modeled *exposure*-based missingness mechanisms are static and the temporal dependencies among items are not utilized, which can naturally influence the subsequent missingness greatly. Consider the following example. If a user has just bought a mobile phone, it is more likely for him/her to buy a suitable phone case next so missingness probabilities of candidate phone cases will be lower than if the user had not bought the phone. Moreover, the effect of item dependencies on the missingness is asymmetric: purchase of a phone case indicates that the user has probably owned a mobile phone and the missingness probabilities of phones should be high during his/her next purchase. Thus the key to modeling the dynamic missingness is how to utilize the temporal information of implicit feedback to capture the asymmetric item dependencies. Instead of finding explicit item dependencies, we assume that the missingness of items for a user at one time is generated by a latent variable called *"user intent"*, and that the dynamics of missingness are driven by a Markov process of *user intents*. In other words, *user intents* capture item relations implicitly and generate time-sensitive *exposure* variables.

Particularly, in this paper, we use a hidden Markov model (HMM) to represent the dynamic missingness of implicit feedback and the estimated missingness of items is incorporated into a probabilistic matrix factorization (MF) model for recommendation. To the best of our knowledge, the proposed framework, namely "H4MF", as a strategy of "leveraging HMM and MF to model the dynamic Missingness for recommendation," is the first work to address the dynamic missingness of implicit feedback in recommendation area. The HMM and MF are seamlessly incorporated in H4MF, making the framework interpretable and extensible. Further, we propose a principled computational algorithm, showing promising results on real-world datasets.

## 2  Related Work

Missing data presents a common challenge for empirical sciences. Most prior studies on recommender systems assumed data is *missing at random* (MAR); however, Marlin and Zemel [2009] demonstrated that data in real recommender systems is not MAR and recommendation algorithms based on MAR assumption may lead to biased results. Several studies have modeled different missingness mechanisms to address the MNAR problem. For explicit feedback, a widely accepted mechanism [Marlin and Zemel, 2009, Ling et al., 2012, Hernández-Lobato et al., 2014] is that missingness is related to the potential ratings (e.g, 1-5 star ratings). Data for items with high ratings are less likely to be missing compared to items with low ratings. For implicit feedback, some causal-process-based methods [Liang et al., 2016, Wang et al., 2018a] first computed exposures for each user and then used them to guide rating prediction, which have shown promising results. Different from these studies, we address the MNAR problem with a dynamic missingness assumption.

Another related work is sequential recommendation, where researchers utilize temporal data for next-item recommendation. Existing sequential recommender systems mainly capture the dynamic user preferences. A popular idea is to utilize Markov chains [He and McAuley, 2016] to model the sequential information. Rendle et al. [2010] proposed a factorized personalized Markov chain (FPMC) model that combines both a common Markov chain and a matrix factorization model. Sahoo et al. [2012] chose a hidden Markov model to capture the dynamic of user preferences for personalized recommendation. However, they did not consider the MNAR problem and the missing data is not well utilized. Some other researchers also used deep learning techniques (e.g., LSTM [Wu et al., 2017] and GRU [Chung et al., 2015] ) for sequential recommendation; however, they are limited in interpretability. In this paper we assume user preferences are static and focus on modeling the dynamic missingness for the MNAR problem. Moreover, it is rather straightforward to extend our framework to capture the dynamic user preferences with existing studies on online learning [Mairal et al., 2010].

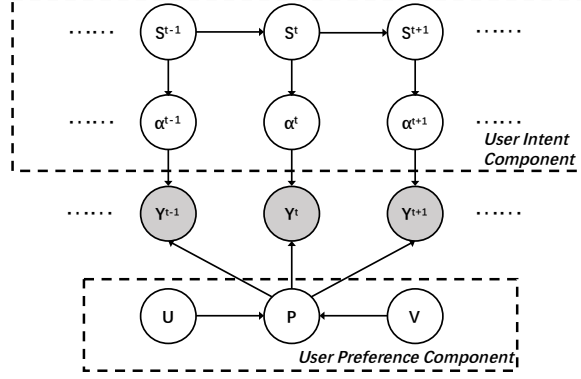

Figure 1: Graphical model of the proposed model.

## 3  H4MF Framework

In this section, we first introduce the problem formula and our proposed framework. Then we describe the parameter inference and the prediction formula in detail.

**Problem Formulation.** Suppose we have $N$ users and $M$ items. For each user $i$, a $T$-length rating history in chronological order is given as $Y_i = \{y_i^1, y_i^2, ..., y_i^T\}$, where $y_i^t$ denotes the item that user $i$ rated at time $t$ (Note that the rating denotes implicit feedback in this paper). The goal of recommender systems is to predict which item the user will rate next, more specifically, $y_i^{T+1}$.

Before describing our model, we first introduce the representation of $y_i^t$ and the definition of missingness variables, which can help to understand the proposed dynamic missingness mechanism. We represent $y_i^t$ as a $M \times 1$ rating vector. As one user can only rate one of $M$ items at one time, there is a "1" in one position of $y_i^t$ and "0" elsewhere. Thus the missing data of implicit feedback refers to "0" entries, which contain negative and unknown feedback. For each $y_{ij}^t$ in dataset, we use a Bernoulli missingness variable $\alpha_{ij}^t$ (same as the *exposure* variable in [Liang et al., 2016]) to indicate the missingness: $\alpha_{ij}^t = 1$ means item $j$ is exposed to user $i$ at time $t$, and $\alpha_{ij}^t = 0$ means the user does not see the item. The missingness variables have a reasonable interpretation: users first have to see the items, then they have the possibility to rate them. Thus $\alpha_{ij}^t$ can be utilized to extract negative feedback from the missing data: if user $i$ has seen item $j$ ($\alpha_{ij}^t = 1$) but the rating $y_{ij}^t$ is 0, this rating is more likely to be negative feedback rather than unknown feedback, which can be further utilized to learn user preference. Note that $\alpha_{ij}^t$ may be different for different $t$ and our model aims to capture its dynamics.

**Model Description.** We assume that user intent and user preference work together for recommendation: User intent determines the missingness of items and user preference determines recommendations from the non-missing items. In this paper we propose a framework named "H4MF" that combines HMM and MF to model the dynamic Missingness for recommendation. As shown in Figure 1, H4MF has two components: the *User Intent Component* and the *User Preference Component*. In the *User Intent Component* we use a first-order hidden Markov model to capture the missingness mechanism. $\alpha^t$ is a $M \times 1$ missingness vector of items at time $t$ generated by a latent state variable $S^t$ (named *"user intent"*), and the probability of $S^t$ depends only on the last state $S^{t-1}$. The *user intent* is a single categorical random variable that can take one of $D$ discrete values, $S^t \in \{1, ..., D\}$. We assume that *user intents* are shared by all users so the generated $\alpha_j^t$ represents $\alpha_{ij}^t$ for all possible users. The state transitions follow a categorical distribution and the conditional observation distribution is defined as:

$$p(y_i^t|S^t, P) = \prod_{j=1}^{M} \sum_{\alpha_{ij}^t} p(y_{ij}^t|\alpha_{ij}^t, P)p(\alpha_{ij}^t|S^t), \ \ \alpha_{ij}^t \in \{0, 1\} \tag{1}$$

In the *User Preference Component*, we adopt a classical but effective matrix factorization model [Mnih and Salakhutdinov, 2008]: the user preference $P \in \mathbb{R}^{N \times M}$ is decomposed as a product of two submatrices $U \in \mathbb{R}^{K \times N}$ and $V \in \mathbb{R}^{K \times M}$, which represent user-specific and item-specific latent

feature factors respectively. More specifically, we use $P_{ij} = U_i^T V_j$ to show the preference of user $i$ toward item $j$. The conditional distribution over the observed ratings $Y_i^t \in \mathbb{R}^{N \times M}$ (the likelihood term) for user $i$ and the prior distribution are given by:

$$p(Y_i|\alpha_i^T, P) = \prod_{t=1}^{T} \prod_{j=1}^{M} [\alpha_{ij}^t \mathcal{N}(y_{ij}^t|P_{ij}, \lambda_y^{-1} I_K) + (1 - \alpha_{ij}^t)\mathbb{I}[y_{ij}^t = 0]],$$

$$p(\alpha_{ij}^t|S^t) = Bernoulli(\mathbb{I}_s(\mu_j^t)), \ \ \mu_j^t \sim Beta(a^t, b^t), \tag{2}$$

$$p(U|\lambda_u) = \prod_{i=1}^{N} \mathcal{N}(U_i|0, \lambda_u^{-1} I_K), \ \ p(V|\lambda_v) = \prod_{j=1}^{M} \mathcal{N}(V_j|0, \lambda_v^{-1} I_K),$$

where $\mathcal{N}(x|\mu, \lambda)$ denotes the Gaussian distribution with mean $\mu$ and precision $\lambda$, $\mathbb{I}[y_{ij}^t = 0]$ is the indicator function that evaluates to 1 when $y_{ij}^t = 0$ is true, and 0 otherwise. $\mathbb{I}_s(\mu_j^t)$ indicates that $\mu_j^t$ is $S^t$-specific. $I_K$ stands for the identity matrix of dimension $K$. $p(Y_i|\alpha_i^T, P)$ can be interpreted as follows: when $\alpha_{ij}^t = 0$, the rating is missing so $y_{ij}^t$ is definitely 0; when $\alpha_{ij}^t = 1$, the rating is not missing so $y_{ij}^t$ is either 0 or 1, depending on the user preference $P_{ij}$. In this paper we present our method and its inference for the case of one user's sequential records; but it is straightforward to apply them to multiple user cases. Note that users have variable-length rating records so that $T$ is not a fixed number for different users.

Next we explain the underlying design of H4MF. We choose HMM for user intent because HMM can well utilize the temporal data to mine the asymmetric item dependencies; and the latent states (*user intents*) can be shared by all users, which simplifies the structure of the missingness mechanism. We choose MF for user preference because MF can model a low dimensional representation for both users and items, which has been proved effective in recommender systems. Meanwhile, H4MF is more explainable and reasonable with this modular structure. Most existing sequential recommendation algorithms [Xiang et al., 2010, Shi et al., 2014] only used "dynamic preference" to account for the temporal user behaviors; they assumed time-varying user preference is the only explanation for the noisy user behaviors. In this case the learned user preference will fluctuate rapidly and be difficult to explain.

Although choosing a dynamic preference model will make H4MF more reasonable, we assume user preference to be static for two main reasons: 1) User preference evolves steadily and is rather stable compared to user intent. Moore et al. [2013] visualized dynamic user preference via trajectories. Their results show that user preferences change steadily and slowly in a long time (month level), especially for older users. In contrast, user intent changes every user-item interaction in H4MF. So it is reasonable to choose static preferences in H4MF. 2) Simplicity for inference is a concern. As our goal to explore the effects of dynamic missingness to recommender systems, MF is also fair for comparison to baselines.

**Parameter Inference.** We choose expectation-maximization (EM) to find the maximum a posteriori (MAP) estimations of the parameters of H4MF. In the E-step, we compute the expected log posterior of the observed data and the user intents, which is:

$$\log p(\alpha_i^T, P|S^T, Y_i) \propto \log p(Y_i|\alpha_i^T, P) + \log p(\alpha_i^T|S^T) + \log p(S^T) + \log p(P) \tag{3}$$

The $\log p(P)$ is computed as $\log p(U|\lambda_u) + \log p(V|\lambda_v)$ and $\log p(\alpha_i^T|S^T)$ is computed as $\log \mathbb{I}_s(\mu_j^t) + \log p(\mathbb{I}_s(\mu_j^t)|a^t, b^t)$; we add a prior to regularize the $\mu_j^t$. As the exact expectation of HMM is computationally intractable, we use Gibbs sampling to infer the posterior probabilities of $S^t$. For a given rating sequence $\{Y_i^t\}$ by user $i$. $S^t$ is sampled from

$$p(S^t|S^{t-1}, S^{t+1}, Y_i, \alpha_i^t, P) \propto p(S^t|S^{t-1})p(S^{t+1}|S^t)p(y_i^t|\alpha_i^t, P), \tag{4}$$

where $p(S^t|S^{t-1})$ and $p(S^{t+1}|S^t)$ can be obtained from the state transition matrix of the HMM, and the expectation of log likelihood of one rating record $y_i^t$ is given by:

$$\log p(y_i^t|\alpha_i^t, P) = \sum_{j=1}^{M} \log \Big( y_{ij}^t \mu_j^t \frac{\mathcal{N}(1|U_i^T V_j, \lambda_y^{-1})}{\mathcal{N}(0|U_i^T V_j, \lambda_y^{-1}) + \mathcal{N}(1|U_i^T V_j, \lambda_y^{-1})}$$

$$+ (1 - y_{ij}^t)(1 - \mu_j^t + \mu_j^t \frac{\mathcal{N}(0|U_i^T V_j, \lambda_y^{-1})}{\mathcal{N}(0|U_i^T V_j, \lambda_y^{-1}) + \mathcal{N}(1|U_i^T V_j, \lambda_y^{-1})})\Big) \tag{5}$$

In the M-step, we maximize the log posterior with respect to $\mu$, $U$, $V$, and $\{S^t\}$. We use gradient ascent to update $\mu$, and compute optimal $U$ and $V$ by setting their derivatives to zero. The details are included in Appendix 1.1. Note that we update $U$ and $V$ and fix the hyperparameters $\lambda_u$, $\lambda_v$, and $\lambda_y$. This strategy follows the original PMF [Mnih and Salakhutdinov, 2008] for simplification. For user intents $\{S^t\}$, we use the Baum-Welch algorithm [Ghahramani and Jordan, 1996] to update the transition matrix and initial states probability distribution of the HMM; as a strict EM-type algorithm it is guaranteed to converge to at least a local maximum.

**Making Prediction.** In the recommendation phase we are interested in the prediction of $y_{ij}^{T+1}$ for user $i$ given his/her previous rating records. We make predictions by integrating out the uncertainty from the missing variable $\alpha_j^{T+1}$:

$$
\begin{aligned}
\mathbb{E}_y[y_{ij}^{T+1}|P_{ij}] &= \mathbb{E}_\alpha\left[\mathbb{E}_y[y_{ij}^{T+1}|\alpha_j^{T+1}, P_{ij}]\right] \\
&= \sum_{\alpha_j^{T+1}\in\{0,1\}} p(\alpha_j^{T+1})\,\mathbb{E}_y[y_{ij}^{T+1}|\alpha_j^{T+1}, U_i, V_j] \\
&= \mu_j^{T+1} \cdot U_i^T V_j
\end{aligned}
\tag{6}
$$

where $\mu_j^{T+1}$ is determined by the next user intent $S^{T+1}$, which can be predicted with the forward algorithm of HMM.

## 4 Further Constraints on Items

Currently all missingness variables $\{\alpha_j^t\}$ share the symmetric Beta priors. One potential drawback is the learned *user intents* may be redundant and items under the same *user intent* tend to have similar missing probabilities. In this section we define two kinds of constraints, namely *inner constraint* and *outer constraint*, to specialize the Beta priors of missingness variables of different items under different *user intents*. The intuitions are simple but reasonable: Items have relations under the same user intent and their *exposure* variables are related. We use the *inner constraint* to denote the influences from other items under the same user intent to one item's missingness. Meanwhile, the missingness of one item under different user intents should follow some patterns to reduce the redundancy. And we use the *outer constraint* to denote the influences from the same item under different user intents to one item's missingness. We adopt a simple implementation: we update the Beta priors in every M-step as follows:

$$
a_{\text{new}}^d \leftarrow a_{\text{ini}}^d + \sigma_j^d \lambda_{\text{Inner}} + \omega_j^d \lambda_{\text{Outer}}, \quad b_{\text{new}}^d \leftarrow b_{\text{ini}}^d + \lambda_{\text{Inner}} + \lambda_{\text{Outer}}, d \in \{1,...,D\}
\tag{7}
$$

Where $a_{\text{ini}}^d$ and $b_{\text{ini}}^d$ are initial Beta priors, $\lambda_{\text{Inner}}$ and $\lambda_{\text{Outer}}$ are the scale parameters, $\sigma_j^d = \frac{\#\text{records of item } j \text{ under user intent } d}{\#\text{total records under user intent } d}$ indicates the occurrence probability of item $j$ with respect to other items under user intent $d$, and $\omega_j^d = \frac{\#\text{records of item } j \text{ under user intent } d}{\#\text{total records of item } j}$ indicates the occurrence probability of item $j$ that is "triggered" by user intent $d$. Then the $a^d$ and $b^d$ are not global constants during the EM procedure and play a constraint role. The items with similar occurrence probabilities under the same user intent will have similar Beta priors. Instead of putting constraints directly on $\mu_j^d$, this strategy can avoid sophisticated inferences and later experiments prove its effectiveness. In experiments we denote this constrained version as H4MF$_c$.

## 5 Experimental Results

In this section we describe the used datasets and experimental settings, evaluate the performance results, and analyze the *user intent* and the item constraints.

### 5.1 Datasets and Settings

We evaluate the performance of our method on three real-world datasets: 1) *MovieLens-100K* dataset ($\sim$ 100 thousand ratings from 943 users on 1,682 movies). The dataset was collected during the seven-month period from September 19th, 1997 through April 22nd, 1998. 2) *MovieLens-1M* dataset ($\sim$ 1 million ratings from 6,040 users on 3,706 movies). The dataset was collected from April 25th,

2000 through February 28th, 2003. 3) *LastFM* dataset ($\sim 100$ thousand ratings from 1,892 users on 17,632 movies). The time period is from August 1st, 2005 through May 1st, 2011. We transform the two *MovieLens* datasets into implicit data by setting ratings that are $\geq 3$ to "1" and the others to "0". We then choose four prevalent methods for comparison, including: (1) PMF [Mnih and Salakhutdinov, 2008], a classical matrix factorization approach that is widely applied as a benchmark. (2) WMF [Hu et al., 2008], a standard matrix factorization model for implicit data, which uses a simple heuristic where all unobserved user-item interactions are equally down weighted against the observed interactions. (3) FPMC [Rendle et al., 2010], a sequential recommendation algorithm based on personalized transition graphs over underlying Markov chains. It used a variant of Bayesian Personalized Ranking (BPR) [Rendle et al., 2009] for optimization. (4) ExpoMF Liang et al. [2016], a probabilistic approach that incorporates user *exposure* to items into collaborative filtering. The baselines are chosen for the following reasons: PMF and FPMC can been seen as sub-models of H4MF, while they overlook the missing data problem. WMF treats the missing data as a MAR problem. ExpoMF takes a static method to the MNAR problem. The main goal of the experiments is to show that how we treat the missing data makes a difference.

We adopt *Hit Ratio* (HR) and *Normalized Discounted Cumulative Gain* (NDCG) to measure the item ranking accuracy of different algorithms. HR measures whether the ground truth item is present on the ranked list, while NDCG measures the ranking quality by considering the positions of hits. We follow the definitions of HR and NDCG in [He et al., 2015]. In our study we always report the averaged HR and NDCG across users. We split the dataset for experiments with the following strategy: we first sort the historical ratings of each user by time order. Then the last records of users are used as test data, the second last records are used as validation data, and the remaining records are used for training. We search for the optimal parameters to maximize the performance on validation data and evaluate the model on test data. For the parameters of baseline models, we refer to their original papers and follow their tuning strategies.

## 5.2 Analysis of Prediction Performance

We report the performance of our methods and baseline models with optimal parameters. For PMF, we set $K = 10$. For WMF, we set $K = 10$, $\alpha = 0.4$. For ExpoMF, we set $\lambda_\theta = 0.01$, $\lambda_\beta = 0.01$, $\lambda_y = 0.01$, and $K = 30$. For our models, we set $\lambda_\theta = 0.1$, $\lambda_\beta = 0.1$, $\lambda_y = 0.1$, $K = 30$, $a^d_{ini} = 1$, and $b^d_{ini} = 2$. For item constraints, $\lambda_{\text{Outer}}$ is set as 1, and $\lambda_{\text{Inner}}$ is set 10, 1, and 0.1 for *MovieLens-100K*, *MovieLens-1M*, and *LastFM*, respectively. We show the performance of our methods with other baseline models in Table 1. As shown in the results, H4MF$_c$ achieves higher item ranking accuracy than the other compared algorithms due to the capability of better capturing the missingness of implicit feedback. Note that PMF, WMF, ExpoMF, and H4MF model user preference similarly: they all use a basic matrix factorization method and the main difference is the way they model the missing data. PMF performs poorly because the datasets are sparse and all the missing entries are treated as negative feedback. So the positive feedback is overwhelmed by negative feedback, leading to a biased user preference learning. FPMC has the same reason for its poor performance. Besides, it is originally proposed for next-basket recommendation. Here we set basket size as 1 as we do not have basket information, which also limits the effectiveness of FPMC. WMF is better than PMF as it treats the missing data with a globally fixed low confidence. ExpoMF models exposure variable $\alpha_{ui}$ for every user-item pair so it can capture more information from the missing data compared to WMF and PMF. H4MF is better than ExpoMF because it considers the dynamic missingness of items. Note that the experimental results of WMF, ExpoMF, and H4MF are very close; WMF even beats WMF and H4MF on *LastFM*. This is because modeling missingness for each missing entry adds model complexity and is prone to overfitting. On the other hand, the superiority of H4MF$_c$ compared to H4MF proves the effectiveness of the user intent constraints.

## 5.3 Analysis of User Intents

This section analyzes user intents in three aspects: recommendation overlaps, sensitivity of user intent number, and interpretation of user intents.

**Recommendation Overlaps.** In H4MF, we use user preference and user intent for recommendation. For a particular user with fixed preference, we sample different user intents and see how different are the recommendation lists. We use the term "recommendation overlap" to denote the ratio of common items in Top-N recommendation lists generated by two different user intents. A large

| Effectiveness of models | | | | | | | |
|---|---|---|---|---|---|---|---|
| Dataset | Metrics | PMF | FPMC | WMF | ExpoMF | H4MF | H4MF$_c$ |
| MovieLens-100K | HR@10 | 0.0031 | 0.0021 | 0.1251 | 0.1230 | 0.1317 | **0.1569** |
| | HR@50 | 0.0296 | 0.0212 | 0.3968 | 0.3478 | 0.3990 | **0.4347** |
| | NDCG@10 | 0.0011 | 0.0007 | 0.0501 | 0.0616 | 0.0583 | **0.0779** |
| | NDCG@50 | 0.0066 | 0.0046 | 0.1203 | 0.1101 | 0.1205 | **0.1367** |
| MovieLens-1M | HR@10 | 0.0021 | 0.0034 | 0.0791 | 0.0801 | 0.0805 | **0.0877** |
| | HR@50 | 0.0093 | 0.0129 | 0.2696 | 0.2808 | 0.2704 | **0.3049** |
| | NDCG@10 | 0.0008 | 0.0087 | 0.0372 | 0.0331 | 0.0408 | **0.0435** |
| | NDCG@50 | 0.0022 | 0.0549 | 0.0800 | 0.0675 | 0.0811 | **0.0897** |
| LastFM | HR@10 | 0.0012 | 0.0021 | 0.0835 | 0.0736 | 0.0799 | **0.0945** |
| | HR@50 | 0.0037 | 0.0360 | 0.2144 | 0.1824 | 0.1980 | **0.2298** |
| | NDCG@10 | 0.0004 | 0.0008 | 0.0432 | 0.0352 | 0.0423 | **0.0495** |
| | NDCG@50 | 0.0009 | 0.0074 | 0.0713 | 0.0575 | 0.0639 | **0.0789** |

Table 1: Performance of different models on three datasets.

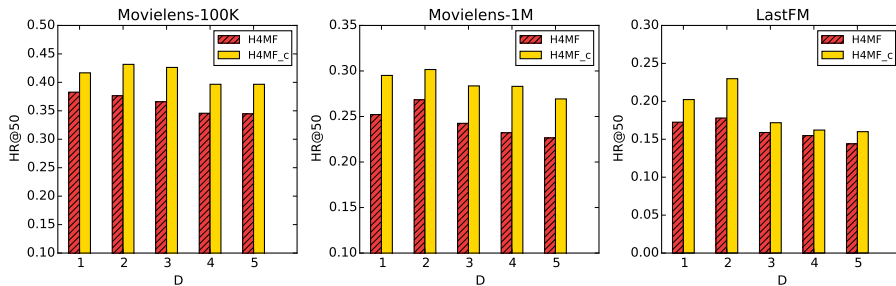

Figure 2: Performances of proposed models with different numbers of user intents ($D$).

recommendation overlap indicates that the two user intents have similar missingness mechanisms. We choose $N = 10$ and show the average of recommendation overlaps across users in Table 2. We can see the recommendation overlaps of H4MF$_c$ are much smaller than those of H4MF, proving that the item constraints can reduce the redundancy of user intents. Meanwhile, the recommendation overlaps decrease both in H4MF and in H4MF$_c$ when $D$ increases. This result conforms to our expectations because our methods can capture more aspects of user intents with a large $D$.

| Recommendation Overlaps of Different User Intents | | | | | | | | | | | | |
|---|---|---|---|---|---|---|---|---|---|---|---|---|
| Dataset | Movielens-100K | | | | Movielens-1M | | | | LastFM | | | |
| User Intent | D=2 | | D=3 | | D=2 | | D=3 | | D=2 | | D=3 | |
| | H4MF | H4MF$_c$ | H4MF | H4MF$_c$ | H4MF | H4MF$_c$ | H4MF | H4MF$_c$ | H4MF | H4MF$_c$ | H4MF | H4MF$_c$ |
| U1 vs U2 | 80% | 26% | 74% | 30% | 92% | 14% | 84% | 6% | 52% | 8% | 52% | 14% |
| U2 vs U3 | - | - | 72% | 18% | - | - | 78% | 2% | - | - | 32% | 2% |
| U1 vs U3 | - | - | 72% | 16% | - | - | 78% | 2% | - | - | 28% | 0% |

Table 2: Recommendation overlaps of different user intents on three datasets. $U1$, $U2$, and $U3$ indicate the indices of user intents. The cases of $D = 4$ and $D = 5$ are included in the Appendix 1.2.

**Sensitivity of User Intent Number.** The number of user intents $D$ is vital to the performance of H4MF. We varied $D$ to train H4MF and presented the prediction results in Figure 2. We can see that H4MF$_c$ performs consistently better than H4MF on all the three datasets. The optimal $D$ is 2 on three datasets. When $D$ increase after $D = 2$, the performance decreases monotonously. Note that in last paragraph we find that the recommendation overlaps decreases when $D$ increase; but this does not guarantee the recommendation performance because a large $D$ will also add model complexity.

**Interpretation of User Intents.** User intents could be utilized to interpret user behaviors and provide explainable recommendations. Table 3 shows a recommendation example of one user in *Movielens-100K* under two different user intents. From the results we can see the genres of recommended movies under *user intent 1* are mainly about *"Crime"* and *"Action"*, but the genres under *user intent 2* are mainly about *"Comedy"*, *"Romance"*, and *"Drama"* (Note that the genre information is not used in model training). Thus we can infer that the user mainly has two tastes in movies. As H4MF

can predict the user's next *user intent*, we will know which genres the user want to see next and provide more precise and interpretable recommendations.

| | User Intent 1 | | User Intent 2 |
| --- | --- | --- | --- |
| Movie Name | Genres | Movie Name | Genres |
| 1. Pulp Fiction | Crime, Drama | 1. Little City | Comedy, Romance |
| 2. Fargo | Crime, Drama, Thriller | 2. The Whole Wide World | Drama |
| 3. Star Wars | Action, Adventure, Sci-Fi, War | 3. Maya Lin: A Strong Clear Vision | Documentary |
| 4. The Full Monty | Comedy | 4. Savage Nights | Drama |
| 5. Contact | Drama, Sci-Fi | 5. Beat the Devil | Comedy, Drama |
| 6. The English Patient | Drama, Romance, War | 6. Ill Gotten Gains | Drama |
| 7. Four Weddings and a Funeral | Comedy, Romance | 7. Withnail and I | Comedy |
| 8. The Fugitive | Action, Thriller | 8. The Inkwell | Comedy, Drama |
| 9. The Princess Bride | Action, Adventure, Romance | 9. Fast, Cheap & Out of Control | Documentary |
| 10. Raiders of the Lost Ark | Action, Adventure | 10. Carrington | Drama, Romance |

Table 3: Top 10 recommendations for one user on *Movielens-100K* under two user intents.

## 5.4 Effectiveness of Item Constraints

To evaluate the effectiveness of item constraints, we tune the $\lambda_{\text{Inner}}$ and $\lambda_{\text{Outer}}$ to observe how they influence the HR@50 of H4MF$_c$. We fix other parameters as described in Section 5.2 and show the results in Figure 3. The optimal parameters are $\lambda_{\text{Inner}} = 10$, $\lambda_{\text{Outer}} = 1$ for *Movielens-100K*, $\lambda_{\text{Inner}} = 1$, $\lambda_{\text{Outer}} = 1$ for *Movielens-1M*, and $\lambda_{\text{Inner}} = 0.1$, $\lambda_{\text{Outer}} = 1$ for *LastFM*. The optimal $\lambda_{\text{Outer}}$ is around 1 for all the three datasets; When it increases, the HR@50 decreases dramatically. Meanwhile, the optimal $\lambda_{\text{Inner}}$ varies across datasets and the performance is less sensitive to the change of $\lambda_{\text{Inner}}$. One main reason is that the total item records under user intents are huge when we have a small $D$. So the ratio measure $\sigma_j^d$ is very small for all items and there are fewer differences among different $\sigma_j^d$, which limits the effectiveness of $\lambda_{\text{Inner}}$. The black dashed lines are the performances of H4MF ($\lambda_{\text{Inner}} = 0$ and $\lambda_{\text{Outer}} = 0$). We can conclude that H4MF$_c$ can achieve improvements with proper constraints, which supports the effectiveness of the two item constraints.

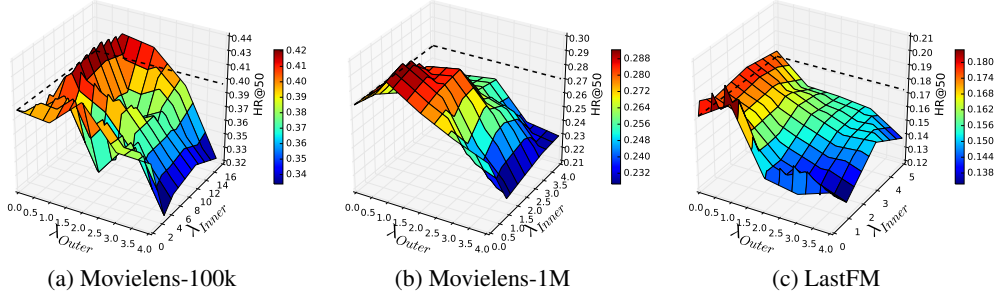

(a) Movielens-100k      (b) Movielens-1M      (c) LastFM

Figure 3: Effectiveness of $\lambda_{\text{Inner}}$ and $\lambda_{\text{Outer}}$ in H4MF$_c$.

## 5.5 Discussions

In this section we first discuss the extensibility and efficiency of H4MF, and then discuss utilization of item relations in recommendation.

**Extensibility.** User intent and user preference can be seen as a factorization of user behavior, which makes H4MF more modular and extensible. We can extend one component without considering the other component. Moreover, both HMM and MF are well studied techniques and their variants can bring insights into H4MF. For example, we can use local low-rank MF [Lee et al., 2013] and mixture-rank matrix approximation [Li et al., 2017] to learn user preference by exploiting the underlying group information of users and items. We can also use hidden semi-Markov model [Yu, 2010] to model the durations of user intents: it is always the case that users purchase serveral items to meet one intent.

**Efficiency.** A potential limitation of H4MF is the time complexity. The inference of the HMM is a bottleneck; its theoretical complexity is $O(\hat{T}D^2)$ for each iteration of the EM method, where $\hat{T}$ is the

length of training data and $D$ is the state number. The experimental runtime results in Appendix 1.3 also reveal that the runtime increases dramatically when $D$ and $\hat{T}$ increase. In real-world applications customers' data are collected accumulatively, so the $\hat{T}$ will become very large. One of the possible extensions is to devise an online version of H4MF. Currently there are several studies related to the online learning of HMM and MF [Mongillo and Deneve, 2008, Mairal et al., 2010], which can be utilized to make H4MF more scalable.

**Item relations in recommendation.** Most recommendation algorithms mainly focus on mining and utilizing the information of item similarity. However, item similarity may lead to meaningless recommendations (e.g., the phone and phone case example in introduction). The key to address this issue is to find asymmetric relations of items. Several researchers [McAuley et al., 2015, Wang et al., 2018b] proposed methods to discriminate substitutes and complements from similar products. But their methods are supervised and the ground truth of labels are directly extracted from user log files, which may contain biases and noise. A more principled approach is to apply techniques of causal discovery to find the directed relations among items. However, current techniques of causal discovery (e.g, modified PC [Spirtes et al., 2000] and GES [Chickering and Meek, 2002]) may not work well on the recommendation data as they are extremely sparse and MNAR. Instead in our model, the asymmetric relations of items are revealed from the temporal data by the dynamical missingness mechanism. In this regard our H4MF can be seen as a step toward causality-based recommendations from similarity-based recommendations.

## 6    Conclusion

In this paper we aim to model and leverage properties of dynamic item missingness to improve recommendation. We proposed a framework that seamlessly combines HMM and MF to model the dynamic missing mechanism of implicit feedback for recommendation. To make the *user intents* less redundant, we introduced two types of constraints for the missingness variables. Empirical results on three datasets show that our method not only outperform alternatives but also provide interpretable recommendations. Further analysis demonstrates the effectiveness of user intent and its constraints. Future work includes extending H4MF with recent advanced variants of HMM and MF.

**Acknowledgments**

This work was supported in part by the National Natural Science Foundation of China (No.U1509221), the National Key TechnologyR&D Program (2015BAH07F01), the Zhejiang Province key R&D program (No.2017C03044). This material is partially based upon work supported by United States Air Force under Contract No. FA8650-17-C-7715, by National Science Foundation under EAGER Grant No. IIS-1829681, and National Institutes of Health under Contract No. NIH-1R01EB022858-01, FAINR01EB022858, NIH-1R01LM012087, NIH-5U54HG008540-02, and FAIN-U54HG008540. Any opinions, findings, and conclusions or recommendations expressed in this material are those of the authors and do not necessarily reflect the views of the United States Air Force or the National Institutes of Health or the National Science Foundation. We appreciate the comments from anonymous reviewers, which helped to improve the paper.

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
