[Supplementary Material]

# 1 Appendix

## 1.1 Update in M-step

For $\mu$, we choose the gradient ascent method. The gradient of $\mu_j^t$ for one rating point $y_{ij}^t$ is:

$$\frac{\partial L}{\partial \mu_j^t} = (1 - y_{ij}^t)\frac{Q_{ij} - 1}{\mu_j^t(Q_{ij} - 1) + 1} + y_{ij}^t\frac{1}{\mu_j^t} + \frac{a^t - 1}{\mu_j^t} - \frac{b^t - 1}{1 - \mu_j^t}, \quad (1)$$

where $Q_{ij} = \frac{\mathcal{N}(0|U_i^T V_j, \lambda_y^{-1})}{\mathcal{N}(0|U_i^T V_j, \lambda_y^{-1}) + \mathcal{N}(1|U_i^T V_j, \lambda_y^{-1})}$. This process is repeated until convergence. For $U$ and $V$, we set their derivatives to zero and get the following update formulas:

$$U_i \leftarrow (\lambda_y \sum_j \bar{\alpha}_{ij} V_j V_j^T + \lambda_V I_K)^{-1}(\sum_j \lambda_y \bar{\alpha}_{ij} y_{ij}^t V_j), \quad (2)$$

$$V_j \leftarrow (\lambda_y \sum_i \bar{\alpha}_{ij} U_i U_i^T + \lambda_U I_K)^{-1}(\sum_i \lambda_y \bar{\alpha}_{ij} y_{ij}^t U_i), \quad (3)$$

where

$$\bar{\alpha}_{ij} = \frac{\bar{\mu}_{ij}\mathcal{N}(0|U_i^T V_j, \lambda_y^{-1})}{\bar{\mu}_{ij}\mathcal{N}(0|U_i^T V_j, \lambda_y^{-1}) + 1 - \bar{\mu}_{ij}},$$

$$\bar{\mu}_{ij} = \begin{cases} \mu_{ij}^t, & \text{if } y_{ij}^t = 1 \\ \sum_{d=1}^{D} \mu_j^d / D, & \text{otherwise} \end{cases} \quad (4)$$

When $y_{ij}^t$ is not rated ($y_{ij}^t = 0$), the $\bar{\mu}_{ij}$ is set as the average of all possible $\mu_j^d$.

## 1.2 Recommendation Overlaps

Below we show the recommendation overlaps when $D = 4$ and $D = 5$.

| | Recommendation Overlaps of Different User Intents | | | | | | | | | | | |
|---|---|---|---|---|---|---|---|---|---|---|---|---|
| Dataset | Movielens-100K | | | | Movielens-1M | | | | LastFM | | | |
| User Intent | D=4 | | D=5 | | D=4 | | D=5 | | D=4 | | D=5 | |
| | H4MF | H4MF$_c$ | H4MF | H4MF$_c$ | H4MF | H4MF$_c$ | H4MF | H4MF$_c$ | H4MF | H4MF$_c$ | H4MF | H4MF$_c$ |
| U1 vs U2 | 78% | 22% | 74% | 10% | 86% | 10% | 86% | 0% | 30% | 4% | 30% | 0% |
| U1 vs U3 | 72% | 44% | 64% | 10% | 84% | 0% | 84% | 0% | 44% | 10% | 38% | 0% |
| U1 vs U4 | 70% | 18% | 66% | 0% | 86% | 8% | 86% | 0% | 28% | 2% | 32% | 4% |
| U2 vs U3 | 66% | 24% | 66% | 20% | 90% | 0% | 82% | 0% | 46% | 6% | 28% | 6% |
| U2 vs U4 | 72% | 16% | 72% | 0% | 90% | 16% | 88% | 8% | 48% | 10% | 24% | 2% |
| U3 vs U4 | 70% | 16% | 64% | 0% | 84% | 0% | 92% | 2% | 32% | 2% | 30% | 0% |
| U1 vs U5 | - | - | 70% | 0% | - | - | 90% | 4% | - | - | 26% | 2% |
| U2 vs U5 | - | - | 72% | 0% | - | - | 88% | 4% | - | - | 32% | 10% |
| U3 vs U5 | - | - | 78% | 0% | - | - | 84% | 2% | - | - | 26% | 2% |
| U4 vs U5 | - | - | 66% | 0% | - | - | 84% | 0% | - | - | 28% | 8% |

Table 1: Recommendation overlaps of different user intents on three datasets when $D = 4$ and $D = 5$. $U1$, $U2$, $U3$, $U4$, and $U5$ indicate the indices of user intents.

## 1.3 Experimental Runtime Results

Below we show the experimental runtime results of H4MF$_c$. We implement the model with Python and our machine settings are listed as follows: Ubuntu 16.04.4 LTS, Intel(R) Xeon(R) CPU E5-2640 v4 @ 2.40GHz, and 12GB 2600 MHz memory. We can see that the runtime is heavily influenced by the state number and the length of the data sequence. When $D$ increases, the runtime increases dramatically. As expected, *Movielens-1M* costs much more time than *Movielens-100K* because users in *Movielens-1M* have longer length of the data sequence.

| Experimental Runtime Results (In Seconds) | | | |
|---|---|---|---|
| State number | MovieLens-100K | MovieLens-1M | LastFM |
| D=1 | 1102 $\pm$ 5 | 23359 $\pm$ 100 | 13639$\pm$ 34 |
| D=2 | 2442 $\pm$ 23 | 36350 $\pm$ 134 | 18268$\pm$ 41 |
| D=3 | 3239 $\pm$ 40 | 42461 $\pm$ 200 | 24694$\pm$ 45 |
| D=4 | 4856$\pm$ 54 | 54461 $\pm$ 389 | 29545$\pm$ 50 |

Table 2: Runtime results of H4MF$_c$.

## 1.4 Notation

| Symbol | Description |
|---|---|
| $y_i^t$ | The item that user $i$ rated at time $t$ |
| $\alpha_{ij}^t$ | Missingness variable of user $i$ toward item $j$ at time $t$ |
| $S^t$ | *User intent* (state) at time $t$ |
| $P_{ij}$ | User preference of user $i$ toward item $j$ |
| $\mu_j^t$ | Prior probability of $S^t$ for item $j$ |
| $a^t, b^t$ | Beta priors of $S^t$ |
| $U_i, V_j$ | User-specific and item-specific latent feature factors |
| $I_K$ | The identity matrix of dimension $K$ |
| $\lambda_u, \lambda_v, \lambda_y$ | Regularization parameters for U, V, Y |
| $\lambda_{\text{Inner}}, \lambda_{\text{Outer}}$ | The scale parameters for update of item constraints |
| $\sigma_j^{S^t}$ | The occurrence probability of item $j$ under $S^t$ |
| $\omega_j^{S^t}$ | The occurrence probability of item $j$ that is "triggered" by $S^t$ |

Table 3: Notation