[Reviews · NeurIPS 2018]

Reviewer 1



This paper proposes using user intents in a dynamic model to address implicit data that is missing not at random for recommendation system problems. The technical ideas of this paper seem sound, as does there model evaluation. I find it particularly compelling that the basic version of the proposed model does not outperform all comparison methods, but that the constrained version does—this shows the importance of the constraints proposed in section 4. The notion of temporal dynamics being important for modeling the missing ness mechanism is interesting. The time between ratings can vary (e.g., the time between T and T+1 is different than the time between T+1 and T+2); unless I misunderstood something, it seems like to proposed omen does not take this into account. Rather, it assumes users are interacting with items evenly across time. I think the irregularity in these interaction intervals is an important piece of information that is currently being ignored. The paper is generally well-written, if a bit dense at times—this seems likely due to page limits. I appreciated the honest discussion of efficiency, the narrative to describe the core model concepts intuitively, and the interpretive results. As a domain expert that focuses on recommendation systems, I would refer to this paper in citations and conversation regularly (I currently reference the exposure model quite a bit). The MNAR problem is very important for recommendation systems with implicit data, but a limited amount of work has been done in this space so far. This is a novel approach to this problem—I doubt that many researchers in this area have considered the relationship between temporal dynamics of users and item as related to the MNAR problem, but the experimental results show that this could be a very important factor.

Reviewer 2



This paper consider the classical problem of matrix completion in collaborative filtering when the data is MNAR -- i.e. with observational data. This problem has already been study (see [1] for instance) and a key problem to handle the bias is to infer the underlying probability of examinations (propensities). Where the previous studies try to predict one static propensity for each user-item pair, this paper proposes to condition these propensities on the sequence of ratings done by the user when such information is available. So they try to model the dynamics of the probability of examination in opposition to other sequential methods such as RNN for reco which aim at modelling the dynamics of the user preferences (when assumed non-stationary). I find the idea of the paper interesting as I feel it answers to a real problem. The proposed approach using a HMM (+MF for scalability) is credible and proposes a good first solution. The experiments are done on popular datasets and prove good performance against the baselines. I especially appreciated the second part of the experimental section where the authors inspect the robustness of their model w.r.t. several parameters (nb of user intent, weight of the constraints). Then, some improvements on the methods would still be required for it to scale to industry-size recommender systems, but the datasets used in experiments are still reasonable. There are still several things that could be improved: - Due to the large amount of notation, the papers is sometimes a bit hard to follow, - The section 4 about constraints on items could benefit from being more developed as it is the part that makes the difference in the experiments - I would be interesting to have more insights on the link with causality from observational data. [1] Modeling user exposure in recommendation, Liang et al.

Reviewer 3



This paper presents H4MF model (HMM + MF for dynamic Missingness) for implicit feedback data. With implicit data, we only observe positive feedback and the missing entries (zeros) in the data can indicate either negative feedback or users are not exposed of the items. H4MF is based on the previous work on modeling user latent exposure (ExpoMF, Liang et al., Modeling user exposure in recommendation, 2016) — the basic idea is that for each user-item pair, there is a latent binary variable to represent exposure; if it’s 1, it means this user is exposed to the item thus 0 feedback mean true negative, while if it’s 0, it means this user have not yet been exposed to this item yet. The difference in H4MF is that H4MF uses a hidden Markov model to capture the temporal dynamics in the user exposure (user intent in this paper). The basic idea is that whether or not a user is exposed to something can be dependent on some other items he/she has been exposed before. The inference is relatively straightforward with some tweaks to overcome the potential redundancy in the learned user latent intents. Empirical results show that H4MF outperforms other similarly-designed baselines. Overall, the paper is clearly written. However, there are a few aspects that I feel this paper falls a little bit short. 1. The idea of adding temporal dynamics in the latent user intent is fairly natural given the flexible model setup in ExpoMF. I like the motivation example about the purchasing phone and phone case (buying a phone will change the exposure to phone case, but conversely buying a phone case will change the exposure to phone in the opposite direction). However, it is not clear to me how the model can achieve this asymmetric reasoning with H4MF — it might be able to achieve it “by accident” but I cannot see how the model “by design” is able to achieve this. I was expecting a qualitative study on real-world data to convince me this but am quite disappointed to not see it. This is also related to the “causality-based recommendation” aspect mentioned in the paper. In the original ExpoMF paper, a major challenge it faces is how to “explain” exposure — there is an example about some user who listens to a lot of music from a certain band but did not listen to one of their songs — we can come up with two equally plausible explanations to this: either the user is not aware of this song, or clearly he knows this song but does not like it, but without the support of additional data, there is no way we can differentiable these two possibilities and establish the link. I don’t see how H4MF can differ from ExpoMF on this front. ########## UPDATED AFTER AUTHOR RESPONSE ########## I want to clarify what I meant by “by accident” as there seems to be some misunderstanding from the author responese. I agree that the temporal HMM model is able to capture the asymmetric relation, my point is more about the model does not have the incentive to capture the motivating example of phone v.s. phone case, which is OK. But in any case, I would be very interested to see a qualitative example demonstrating how H4MF can capture the dynamic missingness in real-world data, similar to the music listening example in ExpoMF. ################################################### 2. Given a temporal user intent model, it seems only natural to also adopt a temporal user preference model — not necessarily deep, but something along the line of dynamic Poisson factorization (Charlin et al., Dynamic Poisson factorization, 2015) or time SVD++ (Koren, Collaborative Filtering with Temporal Dynamics, 2009) is worth trying. I understand the inference with a static MF model is much simpler, but the reason presented in the paper about the preference being more static is not very convincing. 3. The computational cost is quite high and this significantly limits the usability of H4MF — ExpoMF itself is already not computationally friendly, yet H4MF adds a lot more complexity on top of that. The datasets used in the empirically studies are relatively small sized and yet it still takes hours to train the model. I think stochastic algorithm is a promising avenue to overcome this issue. ########## UPDATED AFTER AUTHOR RESPONSE ########## “We implemented ExpoMF in C++ with openMP and it achieved a 30X speedup, compared to the Python implementation.” Please make it publicly available upon publication of this paper. ################################################### Minor comments: 1. Line 8: a essential -> an essential 2. Around Line 40: The phrasing can be improved a little here: is the “missingness probability” referring to non-exposed or true negative? I can roughly infer what the authors mean here from the context, but it would be better to precisely define the term for understanding at first read. 3. Section 4: How is this different from or related to learning the beta prior through empirical Bayes? Updating Beta prior normally mounts to adding the “pseudo-counts” and I don’t think the proposed procedure is clear enough. 4. For implicit feedback data, PMF (Mnih & Salakhutdinov) is not a suitable baseline (as its very subpar results indicate). 5. Maybe time SVD++ or dynamic Poisson factorization is a more suitable baseline than FPMC? 6. Error bars in Table 1? 7. Line 241: Figure 3 -> Figure 2